# Numerical Simulation Research on Mechanical Optimization of a Novel Fastener Type Ballastless Track (NFTBT) for Tram

**Zhiping Zeng** [1,2], **Xiaodong He** [1], **Xudong Huang** [1,*], **Weidong Wang** [1,2], **Di Wang** [1], **Ayoub Abdullah Senan Qahtan** [1], **Weidong Yuan** [1] and **Houdou Saidi Boumedienne** [1]

1   School of Civil Engineering, Central South University, Changsha 410075, China
2   MOE Key Laboratory of Engineering Structures of Heavy Haul Railway, Central South University, Changsha 410075, China
*   Correspondence: xudong_huang@yeah.net

**Abstract:** In view of the problems present in the construction process of the embedded track structure of modern tram, we have designed a Novel Fastener Type Ballastless Track (NFTBT) for tram. To optimize the size of the NFTBT's structure, the finite element model of the NFTBT's structure in the tram running stage is built, the mechanical characteristics of the NFTBT's structure are calculated, and the geometric parameters of the NFTBT's structure are systematically studied. The research results are as follows: (1) As the size of the track slab increases, the displacement differences of the middle part of the NFTBT are similar, and the size of the track slab has little effect on the displacement. (2) The stress difference of the NFTBT's structure under the different distances between the centers of the adjacent grouting holes' conditions is small. Although a larger distance between the centers of the adjacent grouting holes can reduce the number of grouting holes in the track slab, the distance should not be too large to reduce the peak stress at the bottom of the NFTBT. (3) When the distance between the adjacent fasteners of the NFTBT changes within a certain range, the rail is greatly affected by the uneven settlement of the subgrade, and the track irregularity will be aggravated. (4) The NFTBT does not require the implementation of cable passages at intervals, which facilitates the passage and fixation of the cables in the tram operation section and can reduce the difficulty of adjusting the geometry of the track structure, thus accelerating the construction progress.

**Keywords:** tram; novel fastener type ballastless track; finite element; size design; uneven settlement; mechanical characteristics; optimization analysis

## 1. Introduction

Modern tram belongs to the category of urban rail transit, which is similar to subways and light rail transit, but at the same time has its own unique technical and operational characteristics. Modern tram has undergone a comprehensive transformation and upgrade using components of traditional tram, establishing these tram as a new type of public transportation [1–4], as shown in Figure 1. Modern tram can solve the disadvantages of the existing subways, such as expensive investments and long construction stage [5,6]. Based on the old tram system, the modern tram system has been upgraded and modified to reduce noise. The characteristics of low vibration, green environmental protection, low investment, and quick returns comprise the new direction of urban rail transit to the suburbs [7–9].

In view of the complex forms of the ballastless tracks for tram, the weak subgrade, and the unique vehicle type, it is necessary to redesign the under-track infrastructure that is compatible with these ballastless tracks [10–12]. During the construction of the existing embedded track structure, it is difficult to adjust the pouring self-compacting concrete, the geometry of the track structure is not easy to adjust, and the track slab hoisting work is relatively heavy, requiring the cooperation of large construction machinery, especially in urban road sections, which will affect the construction progress [13] (Figure 2).

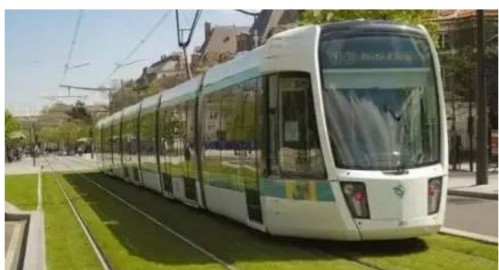

**Figure 1.** Modern tram.

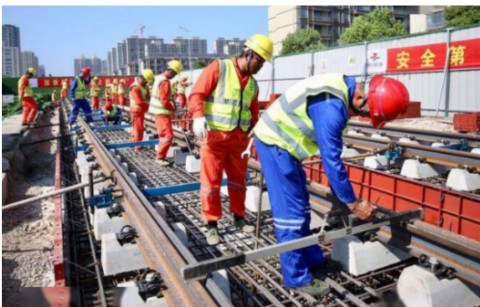

**Figure 2.** Tram track construction site.

Therefore, we have designed a novel fastener type ballastless track (NFTBT) for tram [14], which eliminates the need for spaced cable passage sections, and facilitates the passage and fixation of cables in the tram's operation section. The NFTBT's geometric shape is easily adjusted, which can accelerate the construction progress, as shown in Figure 3. The NFTBT's structure includes a prefabricated track slab body, on which is set a rail seat for fixing groove-shaped rails. The rails are fixed to the rail seat through fasteners to enable the tracks' adjustment and to affix the geometry of the track, which is set at the centerline of the track slab. There are grouting holes and a cable duct laterally penetrating the main body of the prefabricated track slab is provided.

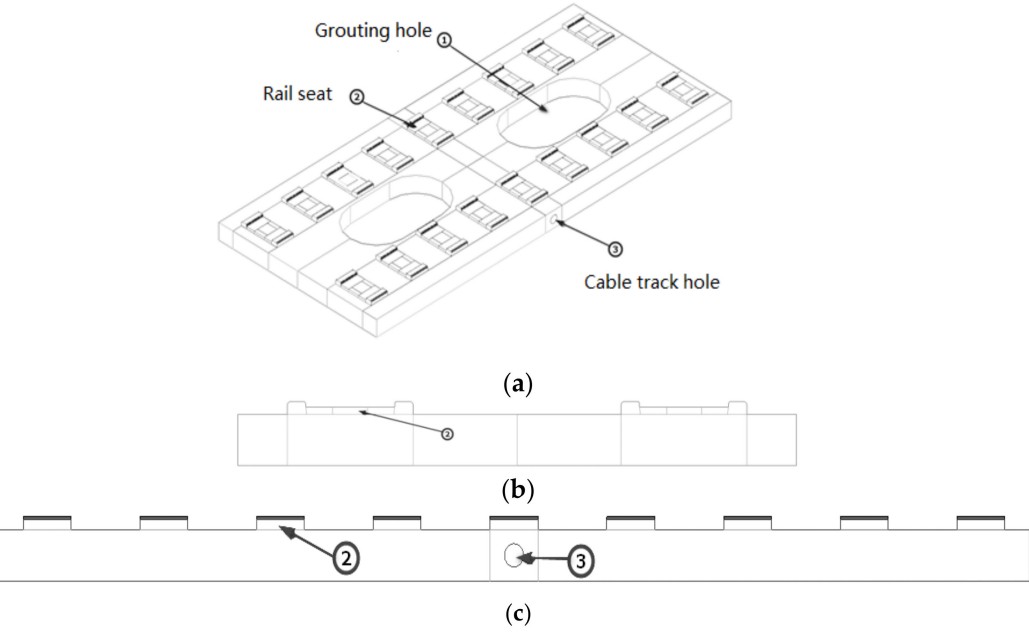

**Figure 3.** Schematic diagram of NFTBT's structure: (**a**) the whole frame; (**b**) schematic diagram of the main view; (**c**) schematic diagram of the side view.

Compared with other types of tram's track structures, the NFTBT has the following advantages:

(1) In the NFTBT's structure, a large area of grouting holes is set at the longitudinal centerline of the track slab. This is conducive to pouring the self-compacting concrete adjustment layer, it facilitates vibrating and compaction, reduces the difficulty of adjusting the geometry of the track structure, and can accelerate the construction progress.

(2) In the NFTBT's structure, a pre-embedded cable channel penetrating the cross-section of the track is set at the center of the track slab, which is beneficial for the cables of the tram's operation section to pass through the line laterally, and there is no need to install cable passage sections at intervals.

(3) The NFTBT can be prefabricated in the factory, which reduces the workload and person-hours of on-site concrete pouring and realizes the goal of environmentally friendly construction.

To study the mechanical properties of the NFTBT's structure during the construction and tram operation stages, it is necessary to select an appropriate mechanical analysis method for research. Since the track structure directly affects the operational safety and stability of modern tram systems [15,16], there is still a lack of research in the existing literature. Gao Liang et al. [17,18] conducted a mechanical analysis on the ladder track structure of the Dalian tram system and put forward optimization suggestions. Chen Peng [19] simulated the wheel–rail dynamic response when a tram passes through a curve. Zhao Wei et al. [20] established a three-dimensional model for the wheel–rail interaction of modern tram. Gao Jiangning [21] introduced the design overview of the track structure of the Shanghai Pudong Zhangjiang tram project. Li Juan et al. [22] took tram slab ballastless track as the research object and analyzed the mechanical properties of track slabs. Wang Shuo [23] mainly used the tram's subgrade structure as the research subject and analyzed the dynamic stress distribution law in the subgrade; Luo Xinwei et al. [24] studied the impact of the key parameters of the rail subgrade on the length of the tram using the orthogonal test method. Li Xianbo [25] analyzed the beam–rail interaction of the seamless track of the tram-embedded track and calculated the allowable temperature of the track.

It can be seen from the existing research and analysis that there are relatively few theoretical studies on tram's track structures, and a large amount of the research conducted by scholars mainly focuses on vehicles, routes, or applicability [26–28]. The research on key technologies is not deep enough to solve many of the problems encountered in design and operation. Therefore, to study the specific force performance of NFTBT, researching the mechanical properties and optimization of NFTBT is not only of great significance for the applicability of the NFTBT but it can also be used as a reference for other types of tram's track structures. Therefore, to reduce the buoyancy of the track caused by the pouring of the cast-in-place concrete adjustment layer, improve the stress state of the precast components, improve the adjustment accuracy of the track geometry, and reduce the difficulty of construction, the NFTBT has been developed, and its structural parameters have been optimized.

## 2. NFTBT Finite Element Model Establishment

### 2.1. Finite Element Model Parameters

This research used ABAQUS software to establish a finite element model of the NFTBT's structure (Figure 4). The ABAQUS software is developed by the Dassault Group in France, and the software version used in this study is ABAQUS 2021, which is released in 2020. The rail is simulated by a beam element, and the filling is simulated by a Cartesian spring element [29]. Other components such as track slabs, concrete adjustment layers, and subgrade support layers are all simulated by solid elements [30]. The rail is considered a long elastic beam, allowing for displacement and corners, and the displacement constraints are given at both ends of the rail; the fasteners have been simplified to point-supported linear springs. Each track slab is designed according to nine groups of fastener nodes, and the rail below the fastener node slab is coupled to a fastener node, then connected to the rail

node with a vertical spring element (Figures 5 and 6). Frictional contact is applied between the track slab and the cast-in-situ concrete adjustment layer, the friction factor is 0.3, and the binding constraint is set between the adjustment layer and the support layer [31,32]. The basic parameters of the track structure selected in the calculation are shown in Table 1.

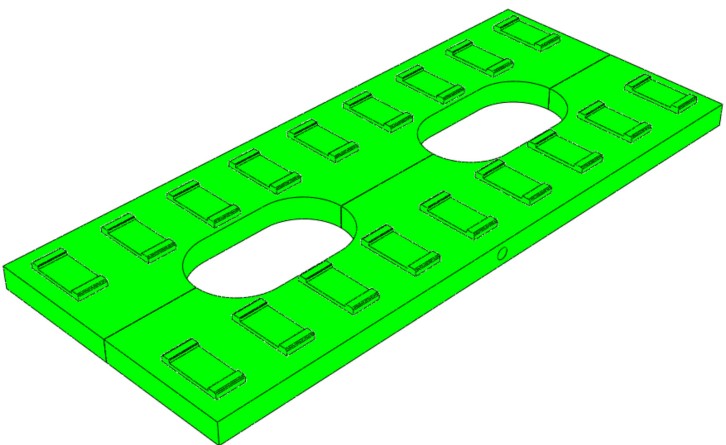

**Figure 4.** Three-dimensional view of the optimized design of the track slab.

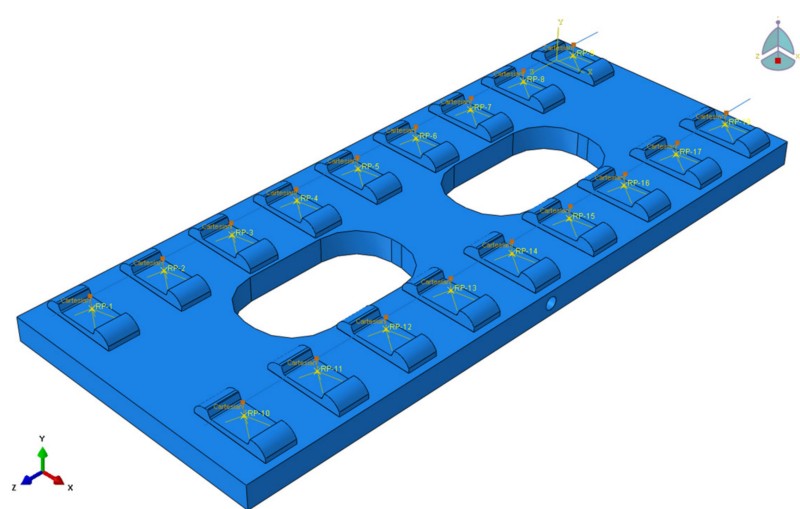

**Figure 5.** The finite element model of the NFTBT's structure.

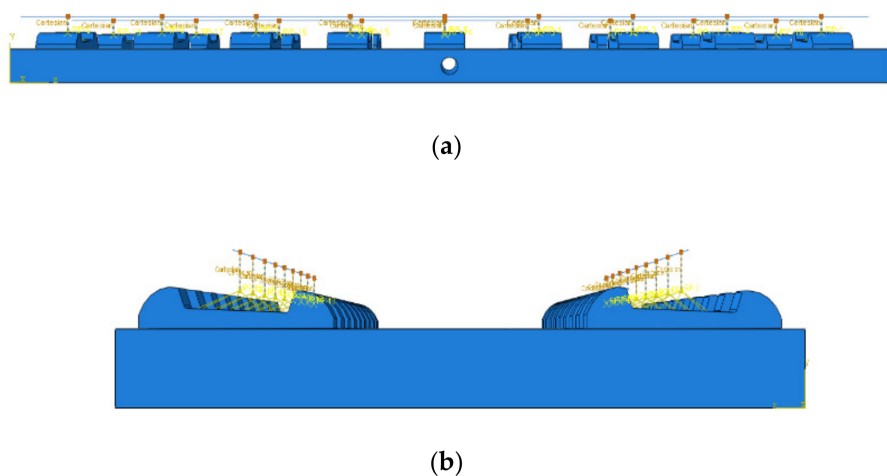

(**a**)

(**b**)

**Figure 6.** The model of rail and fastener system: (**a**) Side view; (**b**) Main view.



**Table 1.** Track structure parameters.

| Track Structure Components | Parameter | Value |
| --- | --- | --- |
| Rail | Type | 50 kg/m |
| | Elastic modulus | $2.1 \times 10^5$ MPa |
| | Poisson's ratio | 0.3 |
| | Linear expansion coefficient | $1.01 \times 10^{-5}$ |
| Wrapping material | Stiffness | 45 kN/mm |
| Track slab | Elastic modulus | $3.45 \times 10^4$ Mpa |
| | Poisson's ratio | 0.167 |
| | Linear expansion coefficient | $1.0 \times 10^{-5}$ |
| Adjustment layer | Elastic modulus | $3.25 \times 10^4$ Mpa |
| | Poisson's ratio | 0.167 |
| | Linear expansion coefficient | $1.0 \times 10^{-5}$ |
| Support layer | Elastic modulus | $3.0 \times 10^4$ Mpa |
| | Poisson's ratio | 0.167 |

During the running stage, the NFTBT's structure mainly bears the load of the tram, and its own weight. In the design process, the additional load of the track structure caused by the uneven settlement of the subgrade should not be included in the design load of the track structure for structural calculation, but it should be regarded as an abnormal load with a small probability of occurrence in the track structure; thus, it can be considered to be a checking condition in the design of the track structure [33]. Considering the influence of the unevenness of the subgrade under the track, the wavelength of the uneven settlement of the track is 20 m, and the amplitude of the uneven settlement is 10 mm, and five track slabs have been selected for modeling and analysis.

The rail is simulated by beam element B31, and the elastic modulus is shown in Table 1. The wrapping material is modeled with the linear spring element CARTESIAN. The track slab, the concrete adjustment layer, and the support layer are all simulated by the solid element C3D20R. The tram load is simulated by multiple concentrated forces and loaded onto the track, and the spacing is set to a fixed wheelbase of 1600 mm. Among them, since the vertical static wheel load is 62.5 kN and the dynamic coefficient is 1.67, the concentrated load value is set to 104.375 kN [34].

Since the uneven settlement of the subgrade is an uneven vertical deformation along the longitudinal direction of the track, there are many forms of settlement of the subgrade. The models for simulating the uneven settlement of the subgrade include cosine, staggered, and angled models [35]. When the ballastless track is located on the roadbed, the uneven settlement of the roadbed will cause the ballastless track to generate additional force. The sine (co)sine settlement curve mainly occurs on the roadbed. Therefore, considering the influence of an uneven settlement on the ballastless track, this study will mainly focus on the sine and cosine-type irregularities. Thus, the commonly used sine-type half-wave curve is selected for this study [36].

$$f = f_0 \sin \frac{\pi x}{l_0} \tag{1}$$

In Formula (1):
$f_0$—limit of uneven settlement amplitude;
$x$—longitudinal coordinate value of the center line;
$l_0$—standard settlement length, which is taken as 20 m.
When the calculated settlement length is not equal to the standard length, the settlement amplitude is calculated according to the following formula:

$$f = \frac{l^2}{l_0^2} f_0 \tag{2}$$

In Formula (2):

*f*—uneven settlement value taken during the calculation;

*l*—length of uneven settlement taken during the calculation.

Assume that the uneven settlement of the roadbed is set at a settlement amplitude of 10 mm and wavelength of 20 m.

Therefore, when simulating the uneven settlement of the subgrade, the two ends of the rail are restrained by the displacement, and the subgrade ground is given a sinusoidal wave of settlement. When the concrete adjustment layer and the track slab are bonded into a whole body, the concrete adjustment layer is considered the structural layer. A contact is set between the layer and the track slab [37]. The lowest settlement point is set at the front axle of the bogie to explore the deformation and mechanical characteristics of the track structure under the uneven settlement of the subgrade [38]. A finite element model of the track structure under the uneven settlement of the subgrade has been established, as shown in Figure 7.

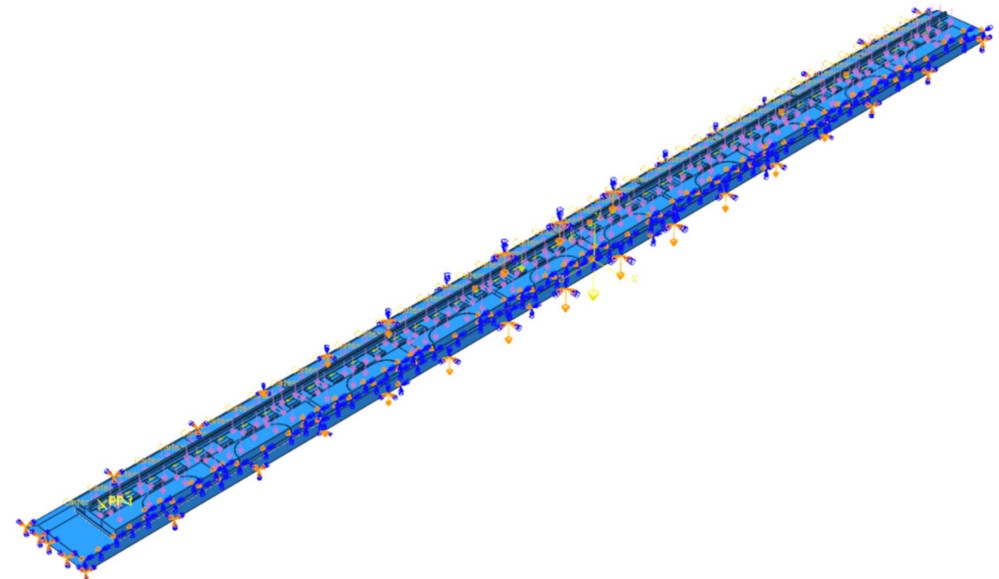

**Figure 7.** Finite element model of track structure under uneven settlement.

### 2.2. Displacement and Stress Measuring Point Layout

The model selects a tram bogie to load onto slab No. 1, selects the deformation and stress output of the track slab, conducts the force analysis under different working conditions, and compares the difference in the deformation and stress of the track structure under different working conditions. The displacement arrangement of the bottom of the track slab below rail seats No. 1–3 is shown in Figure 8, and the location of the track slab and the measuring point of the adjustment layer are shown in Figures 9 and 10, respectively.

The end of slab 3                                      The middle of slab 1

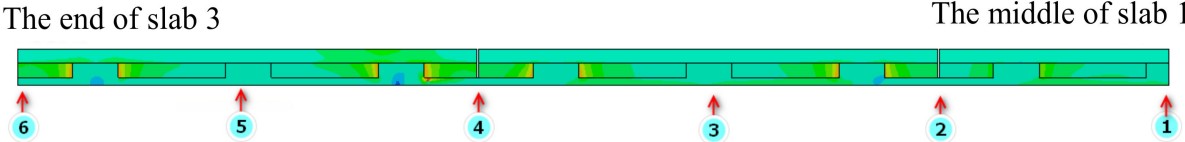

**Figure 8.** Layout of track slab displacement measuring points.

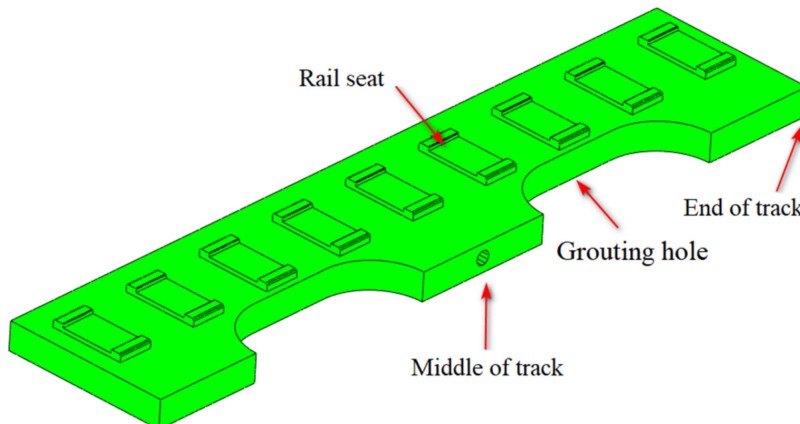

**Figure 9.** Stress measurement position of track slab.

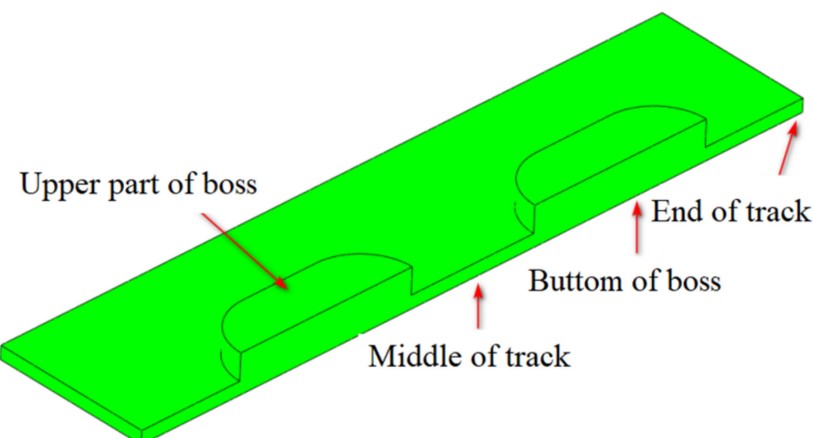

**Figure 10.** Stress measurement position of adjustment layer.

In the operation stage, the NFTBT is mainly subjected to gravity and temperature load. A ballastless track adopts a layered concrete structure to replace the traditional granular gravel bed, while the track slab is prefabricated in the factory with the supporting layer and the adjusting layer poured on site and with different grades of concrete. Due to the difference in the material properties of each rail component, the coordinated deformation of each rack component cannot occur under the influence of an external temperature field. At the same time, the NFTBT boundary is subject to various constraints and cannot form free deformation; thus, the NFTBT produces temperature stress. Combined with the current maintenance experience, temperature deformation may lead to the formation of a disjoint between different NFTBT interfaces and temperature stress may lead to the cracking of the NFTBT.

At this stage, a 1/2 model of the NFTBT's temperature with a gravity load coupled finite element model has been established, as shown in Figure 11, in which the same temperature gradient has been applied to the NFTBT as in the fine adjustment stage of installation with the stress and displacement of the track slab and the concrete of the adjusting layer compared, the stress state and displacement of the interface between the track slab and the concrete of the adjusting layer analyzed, and the interface stress variation trend of NFTBT under positive and negative temperature gradients analyzed so as to predict the interface failure mode.

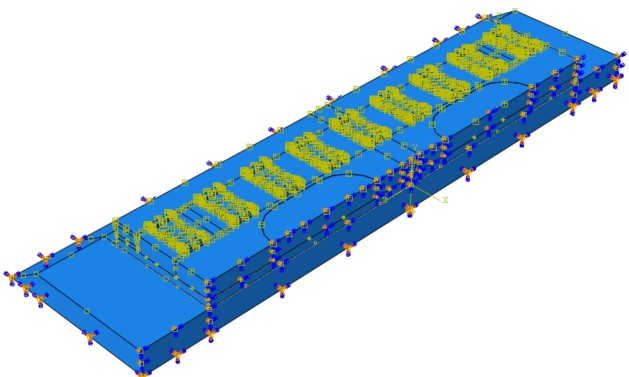

**Figure 11.** Finite element model of 1/2 NFTBT under temperature gradient load.

When the 90 °C positive temperature gradient load and the 45 °C negative temperature gradient load are applied, the initial temperature at the bottom of the track slab is set at 25 °C and 0 °C, respectively, and on the basis of which the linear temperature gradient is applied; thus, the track slab temperature distribution becomes "on the hot under cold," and "on the cold under hot". Since the calculation model only considers the changes on the vertical temperature gradient along the track while the influence of the transverse temperature attenuation and NFTBT temperature transfer along the road is not considered, this is the most unfavorable state for the NFTBT to bear a temperature load. The track slab is bound to the concrete adjustment layer and the bottom of the adjustment layer is bound to the consolidation.

According to the stress and deformation calculation results of the NFTBT at this stage, the temperature deformation law of the NFTBT can be mastered and an optimization scheme for reducing the NFTBT's temperature stress can be put forward.

### 2.3. Working Condition Setting

Through the engineering investigation and analysis, the cable duct radius and passing position of the NFTBT's structure are determined. The cable duct is set to pass through the lateral centerline of the NFTBT, and the pipe radius is set at 80 mm. At the same time, the length, width, thickness, the distance between the centers of the adjacent grouting holes, and the distance between the adjacent fasteners and other parameters of the NFTBT are analyzed.

(1)    Length of the track slab

Since the length of the track slab will change the area of the central connection of the NFTBT, this will result in a local stress concentration of the NFTBT's structure under the action of temperature, the tram, and other loads. It is necessary to optimize the length of the NFTBT to reduce the local force of the NFTBT's structure and optimize the force state of the NFTBT's structure. Four lengths of 4600 mm, 5000 mm, 5400 mm, and 5800 mm are used for the finite element analysis during the running stage. The key parts of the study include the Middle of the track (MOT), the End of the track (EOT), the Grouting Point at the grouting hole (PAGH), the Point under the rail (PUR), the Bottom of the boss (BOB), and the Upper part of the boss (UPOB).

(2)    Width of the track slab

Since the width of the track slab will also change the area of the central connection of the NFTBT, this will result in local stress concentration of the NFTBT's structure under the action of temperature, the tram, and other loads. It is necessary to optimize the width of the NFTBT to reduce the local force of the NFTBT's structure and optimize the force of the NFTBT's structural state. Therefore, four widths of 2300 mm, 2400 mm, 2500 mm, and 2600 mm are used for the finite element analysis of the running stage. The key parts of the study include the Middle of the track (MOT), the End of the track (EOT), the Point at the

grouting hole (PAGH), the Point at the rail seat (PARS), the Point under the rail (PUR),the Bottom of the boss (BOB), and the Upper part of the boss (UPOB).

(3)     Thickness of the track slab

In view of the thickness of NFTBT, considering the locations of embedded pipelines and drainage wells where the line crosses the cable, the finite element analysis of the stress and deformation of the NFTBT's structure during the running stage is carried out in order to improve the stress of the NFTBT's structure, reduce the deformation of the NFTBT's structure, and facilitate its on-site construction. Therefore, four thickness working conditions of 180 mm, 200 mm, 220 mm, and 240 mm are used for research. The key parts of the research include the Middle of track (MOT) in the slab, the End of the track (EOT), the Point at the grouting hole (PAGH), the Point at the rail seat (PARS), the Point under the rail (PUR), the Bottom of the boss (BOB), and the Upper part of the boss (UPOB).

(4)     Distance between the centers of the adjacent grouting holes

Since changing the center distance of the grouting holes will change the area of the central connection of the NFTBT, this will result in the local stress concentration of the track structure under the action of temperature, the tram, and other loads. It is necessary to optimize the grouting hole to reduce the local force of the NFTBT's structure and optimize the NFTBT. The state of the structure is under force. According to the research conclusions of the related track structure's precast components, the grouting hole chamfer radius is 400 mm. Five distances between the centers of the adjacent grouting holes working conditions of 2000 mm, 2200 mm, 2400 mm, 2600 mm, and 2800 mm are used to carry out the mechanical characteristics of the NFTBT's structure during the running stage. For analysis, the key parts of the research include: The Middle of the track (MOT), the End of the track (EOT), the Point at the grouting hole (PAGH), the Point at the rail seat (PARS), the Point under the rail (PUR), the Bottom of the boss (BOB), and the Upper part of the boss (UPOB).

(5)     Distance between the adjacent fasteners

Based on the existing research conclusions, the optimization design of the spacing of NFTBT fasteners is carried out. To reduce the local force of the track structure, optimize the force state of the track structure, and keep the track structure's geometric shape and position stable, four distances between the adjacent fasteners with working conditions of 522 mm, 567 mm, 611 mm, and 656 mm are used for the finite element analysis during the running stage. The key parts of the research include the rail point at the rail (PAR), the point at the rail seat (PARS), the Bottom of the track (BOT), and the Upper part of the track (UPOT).

## 3. Influence of NFTBT Structure's Size Parameters

### 3.1. Influence of Length of Track Slab

After the greening and paving are completed, in the tram running stage, the NFTBT's structure has the combined effect of the tram load, its self-weight, and the paving layer's weight, the track slab displacement under different slab length conditions is shown in Figure 12, and the displacement measurement points are shown in Figure 8.

According to Figure 12, when uneven settlement occurs in the subgrade part of the track structure, under the combined action of the tram load and the uneven settlement, as the slab length increases, the displacement of the middle part of the NFTBT is similar. The surface displacement is different under different slab lengths at the main settlement of the subgrade. However, as the length of the track slab increases, the displacement of the edge of the NFTBT will gradually decrease, and the reduction from the minimum slab length to the maximum slab length can reach 1 mm. The reason for this is that the track's structure is not strong when the slab length is short, which in turn causes the track's geometry to be more affected by the settlement of the roadbed, which is not conducive to maintaining the stability of the track's geometry. At the same time, considering the load diversity of tram

operations, the minimum slab length of the NFTBT should be limited to ensure the stability of tram operations.

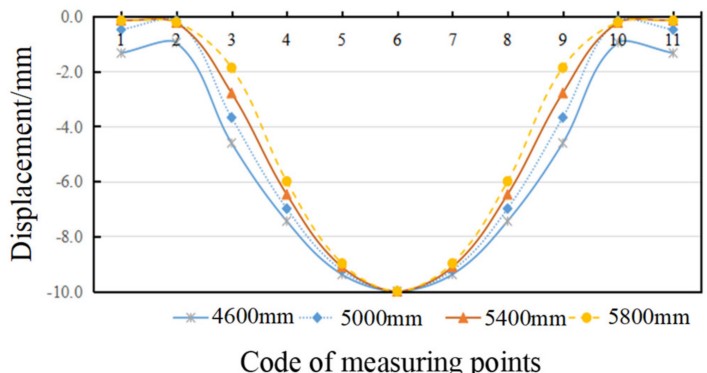

**Figure 12.** Track slab displacement curve under different length conditions.

To compare and analyze the structural stress of the NFTBT under different track slab lengths, according to the key monitoring positions of the track structure determined in Section 2.3, the unfavorable stress at the key positions is analyzed. The stress change curve is shown in Figure 13, which shows:

(1) As the length of the NFTBT increases, the stress at the center of the track slab and the adjustment layer first increases and then decreases, reaching the minimum point at 5400 mm, while the stress at the end of the track slab does not change much. With each slab under long-term conditions, the track slab stress changes little and shows a downward trend, but the stress changes in the adjustment layer are larger and increase further.

(2) With the increase in the length of the NFTBT, the stress at the grouting hole fluctuates greatly, and the stress value at the grouting hole of the track slab and the adjustment layer is the smallest at 5000 mm. In order to achieve the NFTBT's optimal construction, the area of the grouting hole of the track slab must inevitably increase, and the stress state at the boss of the adjustment layer will change from compression to tension. Therefore, the length of the track slab needs to be restricted.

(3) The stress difference in key positions of the NFTBT's structure under different slab length conditions is small (0.25 Mpa), and the stress in the track slab has an opposite trend to the stress of the grouting hole. The stress amplitude of the adjustment layer is smaller than that of the track slab. Since the grouting hole is often a weak link in the construction process, considering the impact of later maintenance, it is recommended that the length of the NFTBT be 5000 mm.

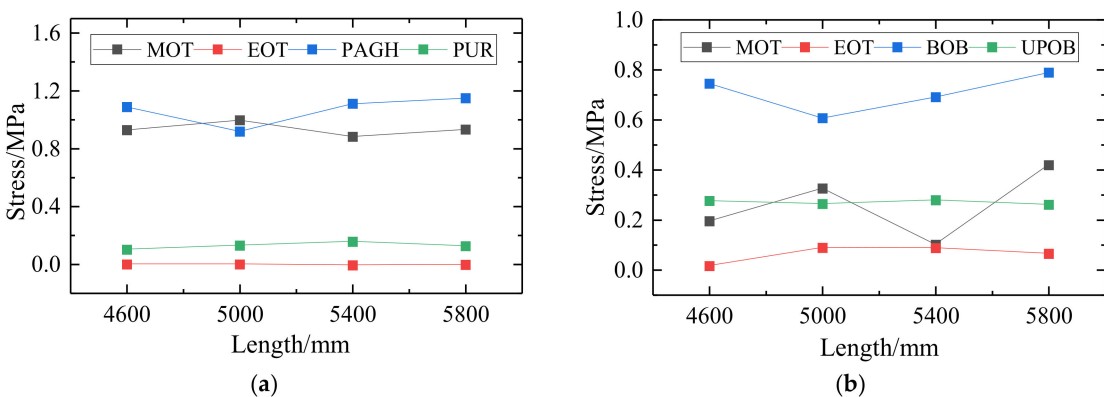

**Figure 13.** Maximum stress change curve of track structure under different length conditions: (**a**) Track slab; (**b**) Adjustment layer.

The stress at the grouting hole of the track slab under a positive temperature gradient load is the control factor of the track slab's design, and the stress in the adjustment layer under a negative temperature gradient is more unfavorable. Therefore, the change in the maximum internal force of the track structure with the length under the temperature load is shown in Figures 14 and 15.

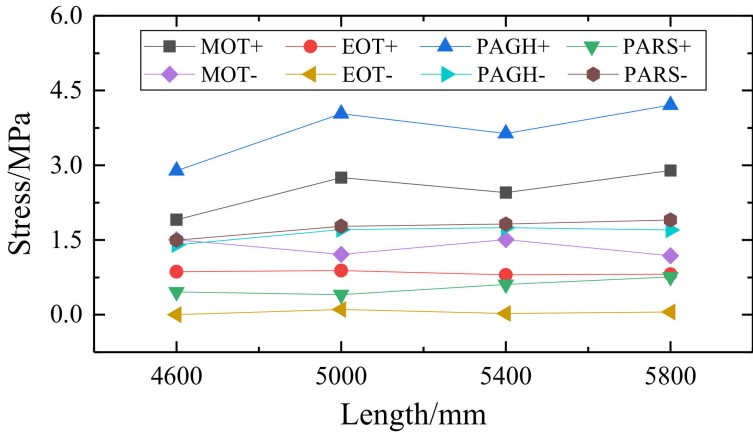

**Figure 14.** Influence of length on temperature stress of track slab (In legend identification, "+" means positive temperature gradient; "−" means negative temperature gradient).

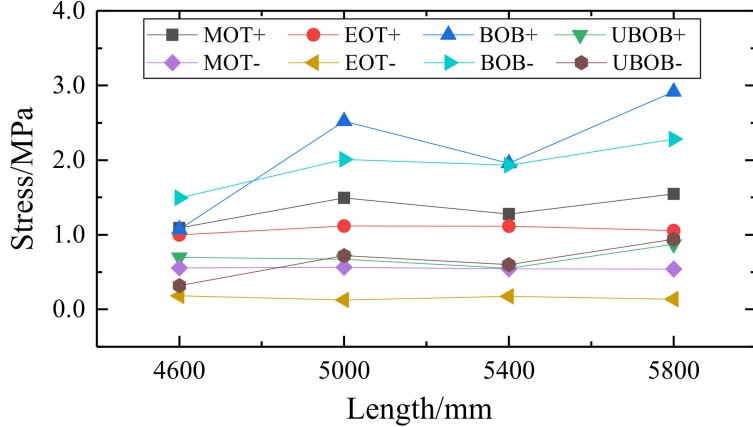

**Figure 15.** Influence of length on temperature stress of adjustment layer (In legend identification, "+" means positive temperature gradient; "−" means negative temperature gradient).

According to Figure 14, the stress of the track slab increases gradually with the increase in the slab length. The stress trend of the slab under negative temperature is opposite, and there is an extreme value at 5000 mm.

According to Figure 15, as the slab length gradually increases, the change trend of stress in the adjustment layer is consistent with that of the track slab. According to the variation law of track structure displacement and stress, the stress states of 4600 mm and 5000 mm at this stage are relatively better.

### 3.2. Influence of Width of Track Slab

After the greening and paving are completed, in the tram running stage, the NFTBT's structure has the combined effect of the tram's load, its self-weight, and the paving layer weight; the displacement of the track slab under different widths is shown in Figure 16, and the displacement measurement points are shown in Figure 6.

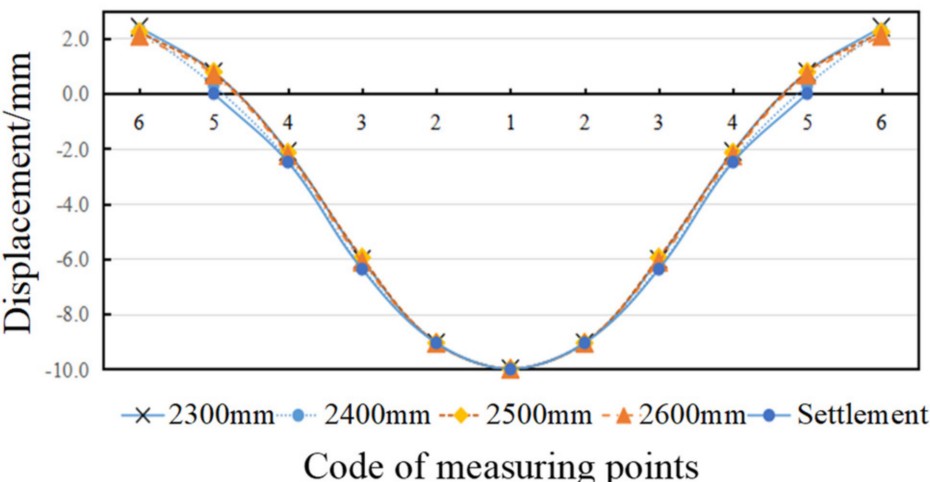

**Figure 16.** Track slab displacement under different width conditions.

According to Figure 16, under the condition of imposing an uneven settlement and the combined action of the tram load and the uneven settlement, as the width increases, the displacement of each point of the track slab gradually decreases, and the track slab displacement change law is basically the same under different working conditions. The displacement at the main settlement position of the roadbed is the largest, and a certain amount of warping occurs in the width of the track slab, but the displacement of each working condition fluctuates slightly with the change in the track slab's width, and thus the track is at the starting point of uneven settlement (near measurement point 5). The maximum difference of slab displacement is 0.5 mm. Combined with the data of measuring points 4~6, the difference in the deformation degree of each track slab is small. It is evident that the width has little effect on the displacement. At this time, in order to control the project cost, it is not necessary to use a large width design.

To compare and analyze the structural stress of the NFTBT under different track slab widths, according to the key monitoring positions of the track structure determined in Section 2.3, the unfavorable stress at the key positions has been analyzed. The main stress change curve in the track slab and the adjustment layer is shown in Figure 17, and it can be concluded that:

(1) In the case of the width of the track slab, the stress changes at each measuring point are not large. Since the grouting hole and the stress in the track slab are the largest, the stress and displacement of the grouting hole and the center of the track are the control factors for the optimal design of the NFTBT at this stage.

(2) As the width of the track slab increases, the stress in the track slab varies the most, showing a trend of first decreasing and then increasing, while the stress at the grouting hole first increases and then decreases very steadily. However, the stress variation of the whole NFTBT's structure is not large (<0.097 MPa)

(3) The stress difference in key positions of the NFTBT's structure under different width conditions is small, and the width of the track slab also affects the overall area and weight of the structure and the occupied land. Considering the stress and deformation of the track structure, uneven settlement, and the conditions under the combined effect of the tram load, in order to facilitate construction and maintenance, the width of the NFTBT should not be too long, so 2400 mm and 2500 mm are recommended as the width for site laying.

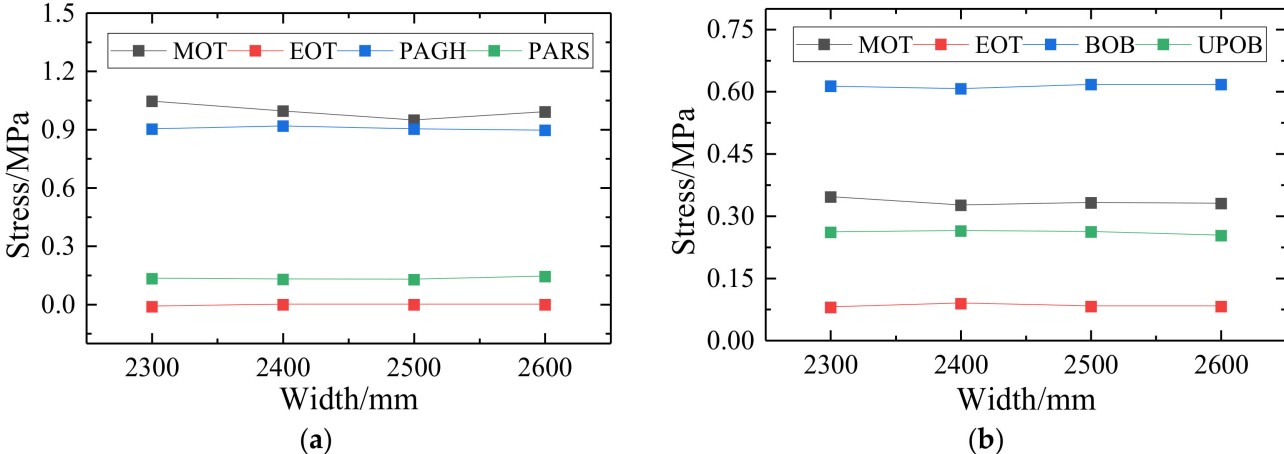

**Figure 17.** Maximum stress change curve of track structure under different width conditions: (**a**) Track slab; (**b**) Adjustment layer.

The change in the maximum internal force of the track structure with the width under the temperature load is shown in Figures 18 and 19.

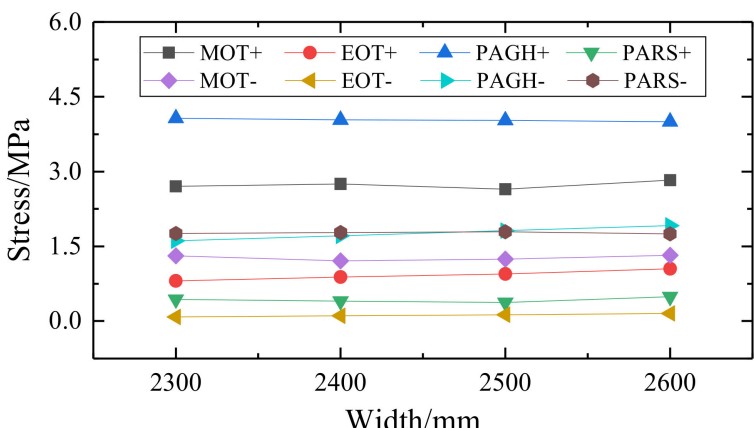

**Figure 18.** Influence of width on temperature stress of track slab (In legend identification, "+" means positive temperature gradient; "−" means negative temperature gradient).

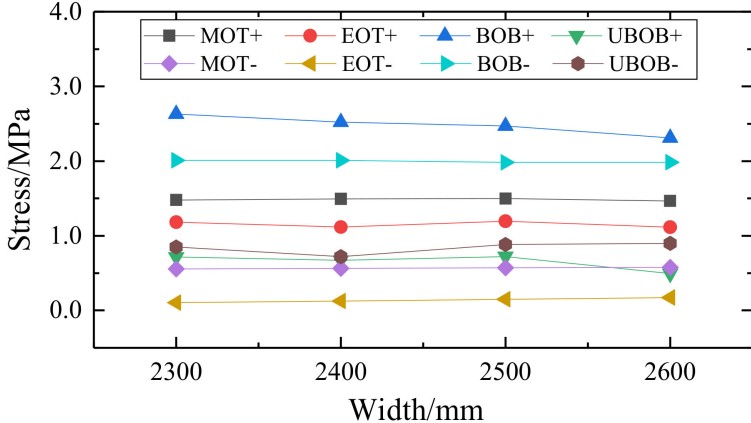

**Figure 19.** Influence of width on temperature stress of adjustment layer (In legend identification, "+" means positive temperature gradient; "−" means negative temperature gradient).

According to Figure 18, the stress fluctuation amplitude of the track slab is smaller than those of the adjustment layer, which are 0.244 MPa and 0.318 Mpa. The stress value



at the grouting hole is large, which should be the control stress for the design of the track structure. The fluctuation of stress and displacement of the track structure under a negative temperature gradient is smaller than that under a positive temperature gradient.

According to Figure 19, the stress amplitude at the bottom of the boss of the adjustment layer is large, which should be fully considered in the subsequent optimization. After the completion of comprehensive laying and before greening, the track structure bears the stress and deformation results of the track slab and adjustment layer under the action of temperature gradient load. The on-site laying scheme shall be determined after a comprehensive comparison of the stress and deformation at each construction stage at 2400 mm and 2500 mm.

### 3.3. Influence of Thickness of Track Slab

After the greening and paving are completed in the tram running stage, when the NFTBT's structure is under the combined effect of the tram load, its self-weight and the paving layer weight (the stress change curve at the unfavorable position of the track slab and adjustment layer under different thickness conditions is shown in the Figure 20), it can be concluded that:

(1) As the thickness of the track slab increases, the self-weight of the NFTBT's structure will gradually increase, which will adversely affect the maximum stress of the track slab. At this time, the stress of the grouting hole increases rapidly, and the stress in the track slab first decreases and then increases. The stress of the boss in the adjustment layer shows an increasing trend, while the stress changes in other positions are smaller.

(2) The stress at the grouting hole and the stress in the track slab are the key control factors for the optimal design of the track structure at this stage. As the slab thickness increases, the local stress at the grouting hole increases and is relatively rapid; the stress in the slab first rises and then drops. There is a minimum turning point in the three working conditions, and both turning points appear near the thickness of 200 mm. At this time, the stress of the adjustment layer is basically unchanged.

(3) Since the thickness of the track slab has a greater impact on construction and economy, it is not the case that the thicker the track slab, the better the overall performance. Synthesizing the law of stress change at each key position, considering the maximum stress at different positions of the track slab, when the thickness of 200 mm is adopted, the stress of the track structure is more favorable than other working conditions.

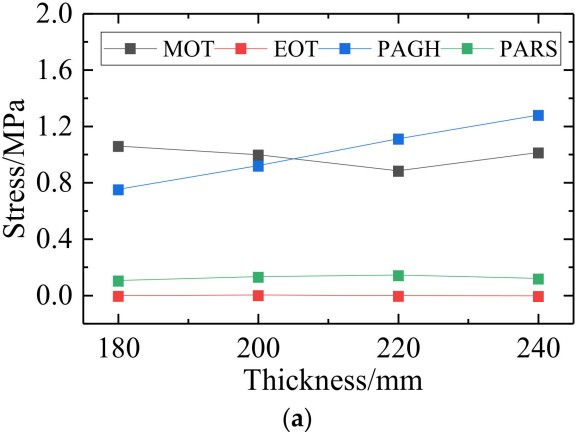
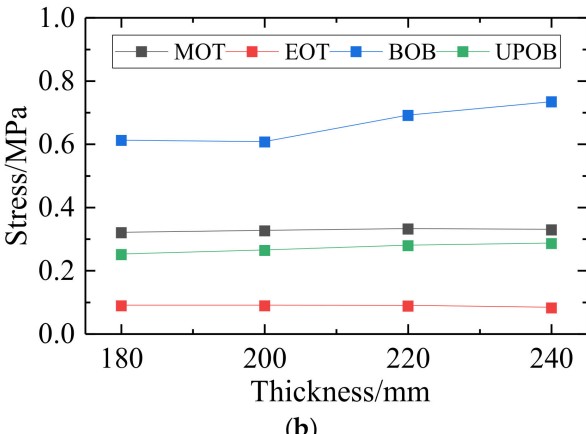

(a)　　　　　　　　　　　　　　　　　　　(b)

**Figure 20.** Maximum stress change curve of track structure under different thickness conditions: (**a**) Track slab; (**b**) Adjustment layer.

The change in the maximum internal force of the track structure with the thickness under the temperature load is shown in Figures 21 and 22.

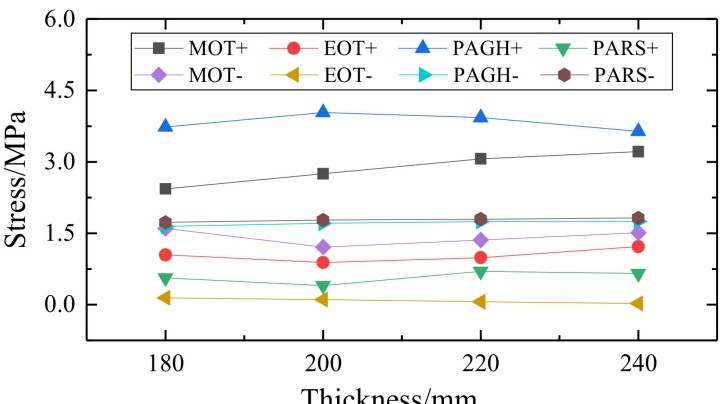

**Figure 21.** Influence of thickness on temperature stress of track slab (In legend identification, "+" means positive temperature gradient; "−" means negative temperature gradient).

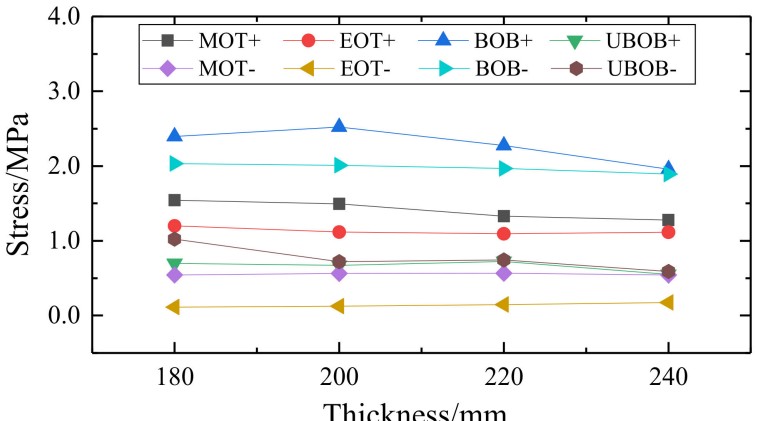

**Figure 22.** Influence of thickness on temperature stress of adjustment layer (In legend identification, "+" means positive temperature gradient; "−"means negative temperature gradient).

According to Figure 21, after pouring the concrete adjustment layer, when the track slab bears the temperature load, the stress at the grouting hole of the track slab is the largest under the positive temperature gradient, so the stress at the grouting hole is the control index of the optimal design. With the increase of the slab thickness, the stress value of the grouting hole first increases and then decreases; on the contrary, there is a turning point at 200 mm at negative temperature, wherein the stress in the slab increases with the increase of slab thickness.

According to Figure 22, the stress of the adjustment layer is more unfavorable under the positive temperature gradient load. The stress at the bottom of the boss first increases and then decreases with the increase of the slab thickness and remains unchanged under the negative temperature gradient. Based on the comparison results of the stress and deformation of the track slab and adjustment layer, the stress condition of the track structure under 200 mm and 220 mm is better before the completion of the greening work.

### 3.4. Influence of Distance between the Centers of Adjacent Grouting Holes

After the greening and paving are completed, during the tram running stage, the NFTBT's structure is under the combined effect of the tram load, its self-weight, and the paving layer weight, as well as the stress on the track slab and the adjustment layer under the conditions of the different distance between the centers of the adjacent grouting holes. The change curve is shown in Figure 23. It is evident that:

(1) The stress difference of the NFTBT's structure under the different distance between the centers of the adjacent grouting holes conditions is small (<0.2 MPa); the stress value

of the NFTBT slab and the grouting hole is larger, and the stress change amplitude of other parts including the rail platform is small. It is evident that the distance between the grouting holes has little effect on the overall force of the track structure.

(2) The stress amplitude of each measuring point at the adjustment layer is smaller than the track slab, and the overall change amplitude is much smaller than the strength value of the concrete. Among them, the stress amplitude at the bottom of the boss is larger, and there is a minimum value when the center distance of the adjacent grouting hole is 2400 mm.

(3) The change in the center distance between the grouting holes has little effect on the stress of the track structure. However, a larger distance between the centers of the adjacent grouting holes can reduce the number of grouting holes in the track slab. For a comparative analysis, it is recommended to set the distance between the centers of the adjacent grouting holes to 2200 mm according to the length of the track slab.

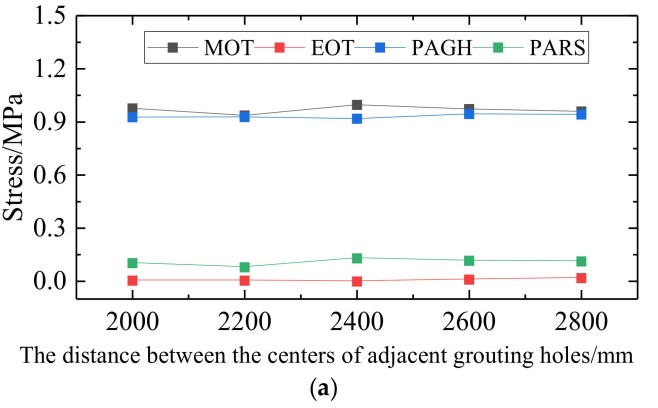 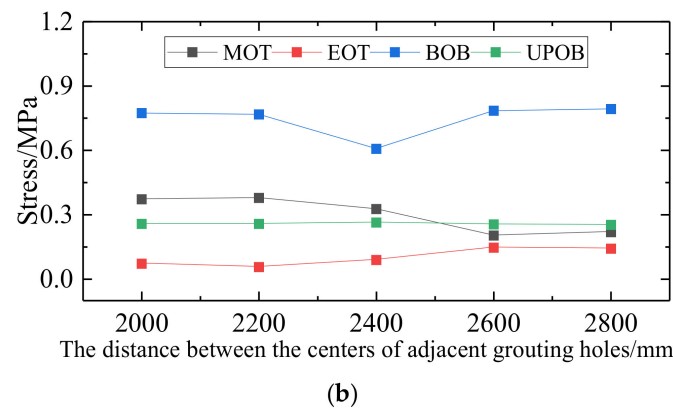

(a)  (b)

**Figure 23.** Stress variation curve of track structure under different distance between the centers of adjacent grouting holes conditions: (**a**) Track slab; (**b**) Adjustment layer.

The change in the maximum internal force of the track structure with the distance between the centers of the adjacent grouting holes under the temperature load is shown in Figures 24 and 25.

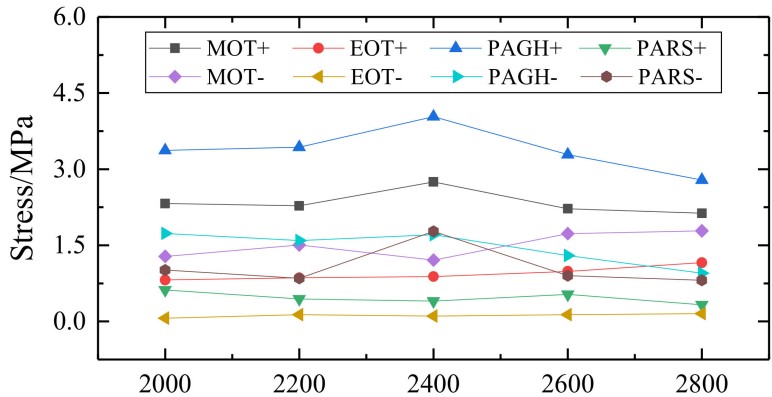

**Figure 24.** Influence of the distance between the centers of adjacent grouting holes on temperature stress of track slab (In legend identification, "+" means positive temperature gradient; "−" means negative temperature gradient).

According to Figure 24, the stress in the track slab and at the grouting hole under the positive temperature gradient increases first and then decreases with the increase of the center distance of the grouting hole. The stress trend of the grouting hole under the negative temperature is the same, but the stress in the slab under the negative temperature

shows the opposite trend. The stress state of the grouting hole of the track slab under the temperature gradient load is relatively unfavorable, and there is a maximum at 2400 mm.

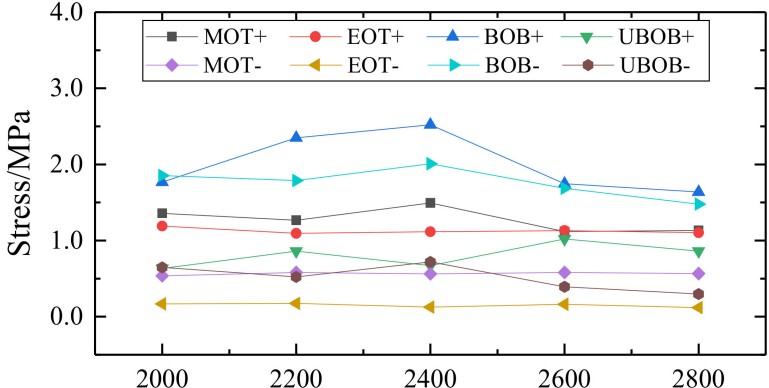

**Figure 25.** Influence of the distance between the centers of adjacent grouting holes on temperature stress of adjustment layer (In legend identification, "+" means positive temperature gradient; "−" means negative temperature gradient).

According to Figure 25, as the center distance of the grouting holes gradually increases, the stress change trend of the adjustment layer is basically consistent with that of the track slab. Under a positive temperature gradient, the stress of the adjustment layer increases first and then decreases, with a maximum value at 2400 mm. Under a negative temperature, the stress change trend at the bottom of the boss is consistent, but the stress in the slab is basically stable. According to the displacement and stress variation law of the track structure, the stress state of 2000 mm and 2800 mm in this stage is relatively excellent.

*3.5. Influence of Distance between Adjacent Fasteners*

Fasteners are important components that connect rails and sleepers or other under-rail foundations, whose function is to maintain the correct position of the rails on the under-rail foundations and the reliable connection between the rails and sleepers. Fasteners can also prevent longitudinal and lateral movement of the rails and provide some elasticity for the NFTBT's structure. Therefore, in the analysis of the influence of the distance between the adjacent fasteners of the NFTBT, the displacement and stress of the NFTBT's structure at the rail seat at the longitudinal centerline of the track slab are mainly used as optimization design indicators to explore the mechanical characteristics of the track structure under the uneven settlement conditions of the tram load and the subgrade.

After the greening and paving are completed, in the tram running stage, when the NFTBT's structure has the combined effect of the tram load, its self-weight, and the paving layer weight, the track structure displacement curve under the different distances between the adjacent fasteners conditions is shown in Figure 26 and the track slab and the stress change curve at the unfavorable position of the adjustment layer are shown in Figure 27.

(1) As the distance between the adjacent fasteners increases, the displacement of the NFTBT rail and rail platform first increases and then decreases, indicating that when the distance between the adjacent fasteners changes within a certain range, the rail is more affected by the uneven settlement of the subgrade, which is due to the larger track irregularity.

(2) As the distance between the adjacent fasteners increases, the rail stress presents an upward trend: the greater the distance between the adjacent fasteners, the more intense the rising trend of rail stress. The stress at the bottom of the track slab first decreases and then increases, indicating that the distance between the adjacent fasteners is at a certain level. When the range is changed, the peak stress at the bottom of the NFTBT can be reduced.

(3) Since the fasteners should have sufficient strength, buckle pressure, proper gauge and level adjustment, good elasticity, and insulation, they should also be standardized as much as possible, simple in structure, and easy to lay and repair. Therefore, the impact of the distance between the adjacent fasteners is related to the overall stiffness of the NFTBT and the ease of construction. Based on the comparative analysis of the displacement and stress changes of the track structure, the recommended distance between the adjacent fasteners is 567 mm.

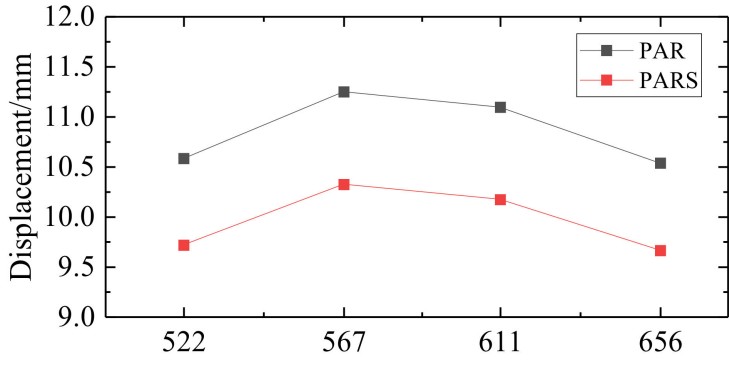

**Figure 26.** Displacement curve of NFTBT's structure under the different distance between adjacent fasteners.

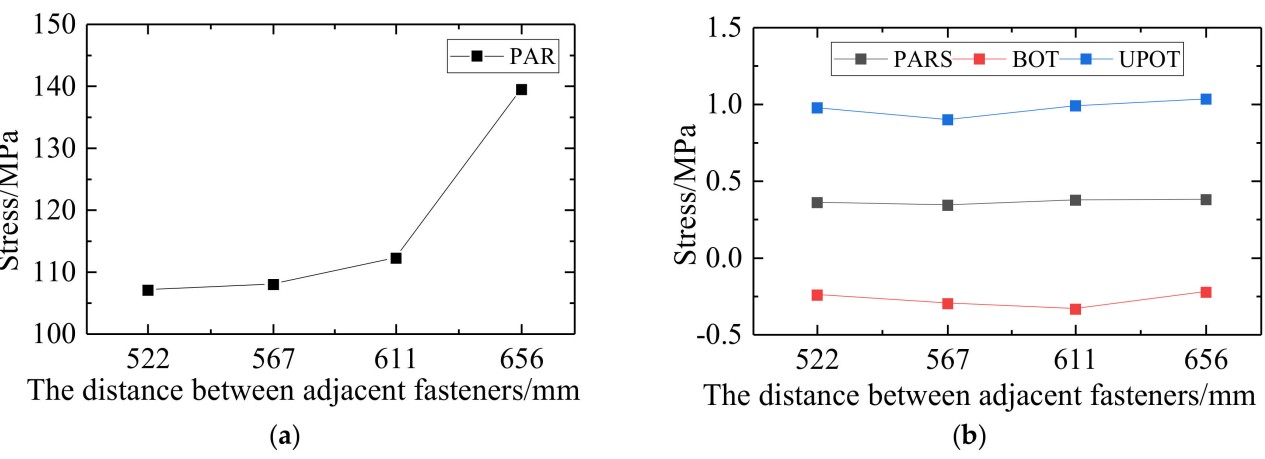

**Figure 27.** Stress change curve of NFTBT's structure under different distance between adjacent fasteners: (**a**) Rail stress; (**b**) Track structure stress.

## 4. Conclusions

The design and application of the novel fastener type ballastless track (NFTBT) for tram is based on the existing problems in the design and construction of traditional subway and tram's track structures and draws on relevant design concepts from all over the world. To optimize the size of the NFTBT's structure, we established the finite element model of the NFTBT's structure in the tram running stage, compared and analyzed different working conditions, and reached the following conclusions:

(1) When uneven settlement occurs in the subgrade part of the track structure, under the combined action of the tram load and the uneven settlement, as the size (length, width, and thickness) of the track slab increases, the displacement of the middle part of the NFTBT is similar, and the track operates under various working conditions. The difference in the degree of slab deformation is small. It is evident that the size of the track slab has little effect on displacement and stress. The size of the track slab affects

the overall area and weight of the structure and its land occupation, as evidenced by the stress and deformation of the track structure. Under the combined action of uniform settlement and tram load to facilitate construction and maintenance, the size of the NFTBT should not be too large.

(2) The stress difference of the NFTBT's structure under the different distances between the centers of the adjacent grouting holes conditions is small, the stress values of the NFTBT slab and grouting hole are larger, and the stress change amplitude of other parts, including the rail platform, is small, showing the distance between the centers of the adjacent grouting holes. The influence on the overall force of the track structure is small. Although a larger distance between the centers of the adjacent grouting holes can reduce the number of grouting holes in the track slab, the center distance of the grouting holes will affect the pouring properties of the adjustment layer, so the spacing should not be too large.

(3) The influence of the distance between the adjacent fasteners is related to the overall stiffness of NFTBT and the convenience of construction. The rail is greatly affected by the uneven settlement of the subgrade, and the track irregularity also increases. As the distance between the adjacent fasteners increases, the rail stress shows an upward trend. The greater the distance between the adjacent fasteners, the more intense the increase in the rail stress; the stress at the bottom of the track slab first decreases and then increases, indicating that the distance between the adjacent fasteners changes within a certain range. At this time, the peak stress at the bottom of the NFTBT can be reduced.

(4) Based on the comparative analysis results of the above working conditions, it is recommended that the NFTBT's structure adopts a unit-slab design, which is composed of rails, fasteners, prefabricated track slabs, adjustment layers, support layers, and other parts. The track slab is a unit-slab structure prefabricated by C60 concrete in the factory. The track slab should have a width of 2400 mm, a thickness of 200 mm, a length of 5000 mm, and a 100 mm expansion joint between the slabs; each track slab should be equipped with two" Semicircle + rectangle + semicircle" grouting holes, the rectangle size should be $800 \times 400$ mm, the circle diameter should be 800 mm, the center distance should be 2200 mm, and the distance between the adjacent fasteners should be 567 mm.

(5) Under the influence of temperature load, the stress of the track slab and adjustment layer increases gradually with the increase in the slab length, and the stress trend in the slab under negative temperature is the opposite; with the increase of the width of the track slab, the stress state of the track slab is more unfavorable, especially at the slab end, which may cause the separation of the contact surface between the self-compacting concrete and the adjustment layer; with the increase of the slab thickness, the stress of the grouting holes first increases and then decreases, whereas under a negative temperature, the stress in the slab increases with the increase of slab thickness; when the track slab bears the temperature load, the stress at the grouting hole of the track slab is the largest under the positive temperature gradient, so the stress at the grouting hole is the control index of the optimal design. Therefore, reasonable parameter settings should be combined with local temperature changes during design.

(6) The NFTBT does not require the implementation of cable passages at intervals, which facilitates the passage and fixation of cables in the tram's operation section, and can reduce the difficulty of adjusting the geometry of the track structure, thus accelerating the construction progress. Therefore, the research regarding the NFTBT is very important for the modern innovation of tram track structures and has a guiding significance. At the same time, it can also provide theoretical references for related tram track structure research.

**Author Contributions:** Conceptualization, Z.Z. and W.W.; methodology, X.H. (Xiaodong He) and X.H. (Xudong Huang); software, D.W.; validation, X.H. (Xiaodong He), D.W. and A.A.S.Q.; formal analysis, X.H. (Xiaodong He), X.H. (Xudong Huang) and W.Y.; investigation, X.H. (Xiaodong He), D.W. and H.S.B.; resources, Z.Z. and W.W.; data curation, X.H. (Xiaodong He), A.A.S.Q. and H.S.B.; writing—original draft preparation, X.H. (Xiaodong He); writing—review and editing, Z.Z. and W.W.; visualization, X.H. (Xiaodong He) and X.H. (Xudong Huang); supervision, Z.Z. and W.W.; project administration, X.H. (Xiaodong He) and X.H. (Xudong Huang); funding acquisition, Z.Z. and W.W. All authors have read and agreed to the published version of the manuscript.

**Funding:** The research reported in this research is partially supported by the Natural Science Foundation of Hunan Province, China, grant number 2019JJ40384; the Science and Technology Development Plan Project of China Railway Bureau 14 Group, grant number YGDC-30.

**Institutional Review Board Statement:** Not applicable.

**Informed Consent Statement:** Not applicable.

**Data Availability Statement:** Not applicable.

**Conflicts of Interest:** The authors declare no conflict of interest.

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
