# Peer review of "Numerical Simulation Research on Mechanical Optimization of a Novel Fastener Type Ballastless Track (NFTBT) for Tram"

_applsci, doi:10.3390/app12178807_

Round 1
Reviewer 1 Report
The article aims to present researches related to the improvement and mechanical characteristics of fasteners for tram. However, the authors present a finite element analysis without experimental validation. Therefore, I suggest to change the title by adding at least the term SIMULATION because the term RESEARCH can create confusions for readers. The article is interesting and could be published after mandatory changes as follows:
Line (34-104): The Introduction section is too long and results of latest researches should be presented in the similar way in all section. The authors use brackets and reference numbers in the first part and other way for the rest of the section (please see Line 105-158, for instance Gao Liang et all). Please presents the state of the art in a harmonized way.
Line 161: Please indicate de version of ABAQUS software, producer, year and country of software company. Do these changes in all document
Line 176: Table 1: I prefer to use 10-5 instead of e-5.
Line 195: please rephrase or check the grammar punctuation
Line 210: please explain for readers what represents x
Line 219: please use the same notations in text: 10mm or 10 mm. In the lines above you used 1600 mm (please see Line 198)
Line 395: Figure 14: In order to have a better view on stress variation at key position of track slab under the influence of temperature gradient, is recommended to have only one graph for both positive and negative temperature gradient. Do the same changes for all graphs where are compared similar parameters. Also, the caption (a) and (b) does not respect the instruction for authors document. It should be bold.
Line 399: Figure 15: same comment as above

Author Response
Dear Reviewer:
Thank you very much for reviewing our manuscript. We have made a comprehensive revision based on your comments and hope to get your approval.
We are very much looking forward to the acceptance of the manuscript. At the same time, if you think that the manuscript still has shortcomings in revision, we hope that you could give us another chance to revise, and we are bound to do our best to complete it.
Thanks again for your guidance!
Sincerely!
Xudong Huang
July 24, 2022

Reviewer 2 Report
A new designed of a ballastless track for tram is proposed, The idea is to eliminate the need for spaced cable passage sections, and facilitate the passage and fixation of cables in the tram operation section.
Finite element model using ABAQUS is performed but the results concern just static analysis considering stress and deformation including the temperature deformation law. The results shows the influence of some parameter on stress and defomations.
The results are interesting and the idea smart.
In my opinion it is important evaluate also dynamic characteristics of ballastless track
What about vibrations? What about niose? Even if the goal of the paper is not dynamic analysis, please argument about this.
Advantages for the construction are described but what does it happen during the use?
Have you compared the results obtained with those systems currently in use?
Can the corrugation forecasted?
Have the authors performed some exeperiments?
Formal presentation has to be revised: tables cross the pages, some captions are too near the figures, figures are not centered...
Author Response
Dear Reviewer:
Thank you very much for reviewing our manuscript. We have made a comprehensive revision based on your comments. For some of your valuable comments, we have explained in detail and hope to get your approval.
We are very much looking forward to the acceptance of the manuscript. At the same time, if you think that the manuscript still has shortcomings in revision, we hope that you could give us another chance to revise, and we are bound to do our best to complete it.
Thanks again for your guidance!
Sincerely!
Xudong Huang
July 24, 2022
